# Functional Suppression of CLOCK Activity in Ventromedial Hypothalamic Prodynorphin Neurons Alters Locomotor Activity and Rapid Eye Movement Sleep

**DOI:** 10.3390/neurolint18010005

**Published:** 2025-12-25

**Authors:** Ting He, Xu Wang

**Affiliations:** MOE Key Laboratory for Biomedical Photonics, Wuhan National Laboratory for Optoelectronics, Huazhong University of Science and Technology, Wuhan 430074, China; ndyfy10030@ncu.edu.cn

**Keywords:** CLOCK activity, REM sleep, circadian rhythms, prodynorphin neurons, ventromedial hypothalamus, sleep–wake regulation

## Abstract

**Background/Objectives**: The circadian regulator, circadian locomotor output cycles kaput (CLOCK), is well-established in maintaining sleep–wake rhythms, yet its cell-type-specific functions in sleep regulation remain largely unexplored. While ventromedial hypothalamic (VMH) prodynorphin (PDYN)-expressing (VMH^PDYN+^) neurons are known to modulate homeostatic and motivational processes, their potential role in circadian sleep regulation has not been investigated. **Methods**: To address this, we developed mice with PDYN neuron-specific functional suppression of CLOCK activity (mClkΔ19) by interfering with their internal clock through Adeno-Associated Virus (AAV)-mediated overexpression of dominant-negative CLOCKΔ19 in PDYN-Cre mice. **Results**: We found that mClkΔ19 mice exhibited reduced locomotor activity during the dark phase, earlier activity peaks, and impaired rhythmicity of rapid eye movement (REM) and non-REM (NREM) sleep. Sleep analysis in mClkΔ19 mice showed selective reductions and fragmentation of light-phase REM sleep, more frequent sleep–wake transitions, and shorter REM cycles during the dark phase, indicating disrupted REM sleep timing. EEG spectral analysis in mClkΔ19 mice revealed decreased gamma activity during REM sleep in the light phase and an increase in delta activity coupled with decreased gamma during wakefulness in the dark phase. **Conclusions**: These findings suggest that the CLOCK activity in VMH^PDYN+^ neurons is vital for circadian accuracy, REM sleep stability, and brain oscillations during sleep–wake cycles.

## 1. Introduction

Sleep is a fundamental physiological process regulated by the interplay between homeostatic and circadian mechanisms [1,2]. Among the various sleep stages, rapid eye movement (REM) sleep plays essential roles in emotional regulation, memory consolidation, and synaptic plasticity [3,4,5,6,7]. The timing, duration, and continuity of REM sleep are tightly governed by the internal circadian clock, which is orchestrated by a core set of genes and proteins, including the central transcription factor *Clock* [8,9,10,11]. While disruption of the *Clock* gene and CLOCK protein has been shown to impair sleep architecture and circadian rhythms at the whole-brain level [12,13,14], its role within specific neuronal populations remains incompletely understood.

Recent evidence has highlighted the hypothalamus, including the ventromedial hypothalamus (VMH), as a potential regulatory node for not only metabolic and defensive functions but also sleep–wake regulation [15,16,17,18,19,20,21,22]. Within this region, neurons expressing prodynorphin (PDYN), a precursor to endogenous opioid peptides, have emerged as modulators of behavioral arousal and circadian-related processes [23,24,25]. Notably, PDYN neurons have been implicated in the sleep–wake cycle [26,27], suggesting that they may contribute to regulating sleep-stage transitions and rhythmicity. However, the contribution of the CLOCK protein within VMH-PDYN neurons to sleep and behavioral rhythms remains unexplored.

To address this question, we employed a cell-type-specific strategy by bilaterally injecting AAV2/1-Ef1α-DIO-EGFP-p2A-mClk-Δ19 into the VMH of PDYN-Cre mice, thereby overexpressing dominant-negative CLOCKΔ19 in PDYN neurons. CLOCKΔ19 is a dominant-negative variant of the CLOCK protein in which deletion of exon 19 eliminates its transcriptional activation function while leaving BMAL1 dimerization intact. When overexpressed, CLOCKΔ19 competitively binds BMAL1 and suppresses endogenous CLOCK-BMAL1-driven transcriptional oscillations. Importantly, this manipulation does not delete or silence the endogenous *Clock* gene; instead, it produces cell-type-specific functional inhibition of the CLOCK protein, allowing us to evaluate the local contribution of CLOCK signaling within PDYN neurons without developmental confounds associated with germline *Clock* mutations. Through 24 h polysomnographic recordings and wheel-running activity monitoring, we found that functional suppression of CLOCK activity in VMH^PDYN+^ neurons attenuated nighttime locomotor activity and altered the circadian distribution of sleep–wake states. This manipulation selectively reduced REM sleep duration, increased REM fragmentation, disturbed sleep–wake transitions, and REM cycling during the dark phase. Moreover, EEG spectral analysis revealed decreased gamma activity during REM sleep in the light phase and increased delta activity coupled with decreased gamma during wakefulness in the dark phase. Together, these findings demonstrate that the CLOCK activity in VMH^PDYN+^ neurons is crucial for maintaining REM sleep quality and coordinating sleep–wake rhythms within the circadian cycle.

## 2. Materials and Methods

### 2.1. Animals

The prodynorphin knock-in mouse line (PDYN-IRES-Cre, JAX #027958), aged 8–10 weeks, was obtained from the Wei Shen Lab (ShanghaiTech University, Shanghai, China) and backcrossed to a C57BL/6J background. All experiments were conducted at the Wuhan National Laboratory under standardized conditions: 22 ± 2 °C, 40–60% humidity, and a 12:12 h light-dark cycle (lights on at 09:00, Zeitgeber Time (ZT) 0). Mice were group-housed with ad libitum access to food and water. All procedures were approved by the Hubei Provincial Animal Care and Use Committee (SYXK2023-0137) and complied with the ethical guidelines of Huazhong University of Science and Technology.

### 2.2. Virus Preparation and Injection

The AAV2/1-Ef1α-DIO-EGFP-p2A-mClkΔ19 construct was a generous gift from the Erquan Zhang Laboratory (NIBS, Beijing, China). Because the plasmid was not designed or cloned in our laboratory, the exact nucleotide sequence is not available. The cassette expresses the mouse CLOCKΔ19 dominant-negative protein and includes an Ef1α promoter, a Cre-dependent DIO inversion system, EGFP, a p2A self-cleaving peptide, and the mClkΔ19 coding sequence. A schematic illustration of the vector design is provided in Appendix A. The mClkΔ19 sequence used in this study encodes the mouse CLOCKΔ19 dominant-negative protein, which lacks exon 19 and competitively inhibits CLOCK-BMAL1 transcriptional activity. Thus, AAV-mediated expression of mClkΔ19 functionally suppresses CLOCK activity without altering endogenous *Clock* gene expression. AAV9-Ef1α-DIO-EYFP was purchased from Braincase Bioscience (Shenzhen, China) for controls. All viruses were stored in aliquots at −80 °C until use. Mice were fixed in a stereotaxic injection frame (Narishige Scientific Instrument Lab, Tokyo, Japan) after receiving an i.p. injection of pentobarbital (50 mg/kg) for anesthesia. Viral injections were performed using the following stereotaxic coordinates: bilateral VMH, −1.46 mm from bregma, ± 0.3 mm lateral from the midline, and 5.4 mm vertical from the cortical surface. After surgery, the animals were kept on a heating pad and relieved of pain by injecting (0.1 mL/10 g) tolfenamic acid injectable solution (125 μL in 10 mL of 0.9% NaCl solution) until they revived. All AAVs were injected at a dose of 0.2 μL and were adequately expressed for at least 4 weeks prior to the experiments. Viral injections were targeted to the VMH; however, occasional spread to nearby hypothalamic regions was observed. Analysis focused on VMH-localized neurons, and EGFP-only controls were included to account for any off-target expression.

### 2.3. Electrode Implantation

To monitor neural and muscular activity, mice were surgically implanted with EEG and EMG electrodes after viral injection. The EEG recording electrode was positioned at stereotaxic coordinates (AP, +1.75 mm; ML, −0.4 mm) relative to bregma, with a reference electrode placed over the cerebellum. Two pre-attached EMG wire electrodes connected to a four-pin connector were bilaterally inserted into the neck musculature. All electrodes were securely anchored to the skull using Super-Bond C&B dental adhesive mixed with acrylic cement. After allowing ample time for viral expression, the animals were connected to flexible recording cables and acclimated to the setup for at least 48 h before electrophysiological data were acquired.

### 2.4. Polysomnographic Recordings

For polysomnographic recordings, the phases of the data were classified based on the characteristics of the EEG and EMG, including NREM sleep, REM sleep, and wakefulness, as detailed in the previous study [28]. The proportion of time spent in each vigilance state was determined by dividing the total duration of that state by the 24 h recording period. For analyses within specific phases (light and dark), the proportion was calculated by dividing the total duration of each state by the corresponding 12 h recording period. The periodicity of REM sleep can be assessed by analyzing the intervals and frequency of REM sleep occurrences. The inter-REM interval (IRI) is calculated as the time difference between the onset of one REM episode and the onset of the next. The mean REM cycle length is determined by dividing the total sum of IRIs by the total number of REM episodes. The REM bout frequency is calculated as the total number of REM episodes divided by the total recording time (hours). Circadian rhythm analysis is assessed by calculating the circadian index, which is determined by dividing the time spent in the light phase by the time spent in the dark phase [29].

### 2.5. EEG Power Normalization

To evaluate spectral dynamics across extended recordings, band-specific power was normalized using a global-mean normalization. Briefly, the EEG signal was first subjected to Welch’s power spectral density estimation (2 s windows, 50% overlap, 0–100 Hz range, 0.2 Hz resolution). For each frequency band of interest (delta (δ): 0.5–4 Hz, theta (θ): 6–10 Hz, sigma (σ): 12–16 Hz, gamma (γ): 30–60 Hz), the band power was computed as the integral of the power spectrum within the corresponding frequency range.

To obtain normalized band energy, the power value of each band at a given recording segment (e.g., hourly data) was divided by the mean power of that band across the entire recording period:$$E_{\mathrm{norm}}(t,b)=\frac{E(t,b)}{\bar{E(b)}}$$
where E(t,b) represents the raw power of band *b* at time segment *t*, and $\bar{E(b)}$ is the average power of that band over all segments. These normalization values reflect relative variations in spectral energy independent of absolute amplitude differences, allowing comparison across time and frequency bands [30,31,32].

### 2.6. Locomotor Activity Recordings

Locomotor activity was monitored using a custom-built running wheel system consisting of an acrylic flying-saucer wheel embedded with four magnets, a Hall sensor, and a transparent housing cage. The signal was processed through a circuit board designed by our laboratory and manufactured by Shenzhen Future Factory (Shenzhen, China). Data were collected with an NI-6009 acquisition card (National Instruments, Austin, TX, USA) and Chart software (Chart V5.1.3, University of Strathclyde, Glasgow, UK). To reduce novelty stress, mice were allowed free access to a wheel in their home cages for 5–7 days before recordings. During the experiment, animals were singly housed for 3 days of acclimation, after which wheel-running activity was recorded continuously for 24 h from the onset of the light cycle. Output files were converted into MATLAB format (MATLAB 2020a, MathWorks, Natick, MA, USA) for analysis, and locomotor distance was calculated from the number of wheel revolutions.

### 2.7. Immunohistochemical Analysis and Microscopy Imaging

Mice were deeply anesthetized and perfused intracardially with 1% phosphate-buffered saline (PBS), followed by 4% paraformaldehyde (PFA) in PBS. Following perfusion, the brains were removed and post-fixed in 4% PFA at 4 °C overnight. They were then cryoprotected in 30% sucrose in PBS and frozen by immersion in 2-methylbutane, which had been chilled to −80 °C. Subsequently, coronal sections (40 μm) were collected using a cryostat (CryoStar NX50,Thermo Fisher Scientific, Eindhoven, The Netherlands). To clarify the virus expression, after being washed with 1% PBS, the brain slices were mounted and stored in an antifade mounting medium with DAPI dihydrochloride at 4 °C until use. Fluorescent images from brain tissue were acquired by a Leica DM4B microscope (Leica Microsystems, Wetzlar, Germany). We used a 10× Plan Apochromat air objective, two laser wavelengths (405 nm and 488 nm), and differential interference contrast (DIC). Image acquisition was controlled by Leica Application Suite X (LAS X 3.0) software.

### 2.8. Quantification and Statistical Analysis

All statistical analyses were conducted using MATLAB 2014b and GraphPad Prism v9. Experimental conditions were assigned randomly, and behavioral data were analyzed following a double-blind protocol. Statistical comparisons utilized unpaired *t*-tests, two-way ANOVA with Sidak’s or Bonferroni’s multiple comparisons test, and the Mann–Whitney U test. Statistical significance was denoted as follows: * *p* < 0.05, ** *p* < 0.01, *** *p* < 0.001, and **** *p* < 0.0001. Refer to the figure legends and Appendix A.

## 3. Results

### 3.1. Functional Suppression of CLOCK Activity in VMH^PDYN+^ Neurons Attenuates Dark-Phase Locomotion and Affects the Circadian Rhythm of the Sleep–Wake Cycle

To examine the role of circadian rhythms in locomotion and sleep–wake cycles regulated by PDYN neurons in the VMH (VMH^PDYN+^), we bilaterally injected AAV2/1-Ef1α-DIO-EGFP-p2A-mClk-Δ19 into the VMH of PDYN-Cre mice. Four weeks after viral expression, we obtained 24 h locomotor activity and sleep recordings, using mice overexpressing dominant-negative CLOCKΔ19 (mClkΔ19) as the experimental group and EYFP-expressing mice (EYFP) as controls (Figure 1A,B). Although viral expression was primarily restricted to the VMH, a small number of EGFP-positive cells were occasionally observed in adjacent hypothalamic regions (Figure 1B, arrows). These off-target cells were sparse and did not appear to influence the behavioral results, as confirmed by the EYFP-only control group.

Compared to EYFP controls, the mClkΔ19 mice showed a significant reduction in locomotor activity during the dark phase (F (23, 264) = 1.873, *p* = 0.011, two-way ANOVA with Šídák’s multiple comparisons test; Figure 1C), although total 24 h activity levels were similar between groups (t (11) = 1.173, *p* = 0.266, unpaired *t*-test; Figure 1D). Additionally, mClkΔ19 mice displayed a phase-advanced activity peak (t (11) = 2.442, *p* = 0.033, unpaired *t*-test; Figure 1E). No significant differences were detected in total running wheel activity or in analyses of the 12 h light and dark phases (Figure 1F,G). However, circadian binning analysis indicated that this decrease was especially notable during the last 6 h of the dark phase in mClkΔ19 mice (*p* = 0.035, U = 6, Mann–Whitney test; Figure 1H). In addition, we also recorded EEG/EMG of both groups; circadian rhythm analysis of sleep stages showed that mClkΔ19 mice exhibited a significant decreasing trend in rhythm indices for both REM (t (12) = 2.120, *p* = 0.056, unpaired *t*-test; Figure 1I) and NREM sleep (t (12) = 2.296, *p* = 0.041, unpaired *t*-test; Figure 1J), while displaying higher wakefulness rhythm indices compared to EYFP controls (t (12) = 2.447, *p* = 0.031, unpaired *t*-test; Figure 1K). These findings demonstrate that suppression of CLOCK activity in VMH^PDYN+^ neurons partly impairs the timing precision of behavioral and sleep rhythms.

### 3.2. Functional Suppression of CLOCK Activity in VMH^PDYN+^ Neurons Reduces the Duration of REM Sleep During the Light Phase

To explore the mechanisms underlying these alterations following suppression of CLOCK activity in VMH^PDYN+^ neurons, we analyzed the duration of each sleep–wake state. We found that mClkΔ19 mice exhibited abnormal changes in REM sleep at specific time points (F (23, 288) = 1.912, *p* = 0.0074; ZT21: *p* = 0.041; two-way ANOVA with Bonferroni’s multiple comparisons; Figure 2A). Notably, compared to EYFP mice, the PDYN neuron-specific *Clock* gene mutation only caused a significant decrease in REM sleep duration during the light phase in mClkΔ19 mice (t (12) = 4.156, *p* = 0.0013, EYFP: 259.1 ± 6.993 s, mClkΔ19: 216.6 ± 7.257 s, unpaired *t*-test; Figure 2B,D). However, no significant differences were observed in NREM sleep or wakefulness duration (wake: F (23, 288) = 1.457, *p* = 0.084; NREM: F (23, 288) = 1.326, *p* = 0.149; two-way ANOVA with Bonferroni’s multiple comparisons; Figure 2E,I), consistent across analyses of total and phase-specific durations (Figure 2F–H,J–L). These results suggest that the disruption of the CLOCK activity in VMH^PDYN+^ neurons selectively impairs the temporal regulation of REM sleep, particularly during the light phase, without significantly affecting NREM sleep or wakefulness. This finding indicates a state-specific role of CLOCK in fine-tuning REM sleep architecture within the circadian cycle.

### 3.3. Functional Suppression of CLOCK Activity in VMH^PDYN+^ Neurons Induces REM Sleep Fragmentation and Irregular Sleep–Wake Transitions During the Dark Phase

To better understand how dominant-negative suppression of CLOCK activity in VMH^PDYN+^ neurons affects REM sleep architecture, we analyzed sleep patterns and transitions in mice. Compared with EYFP controls, we found a significant increase in short REM sleep episodes (<30 s) and a notable decrease in long REM sleep episodes (>120 s) in mClkΔ19 mice (F (6, 84) = 7.271, *p* < 0.0001, two-way ANOVA with Šídák’s multiple comparisons test, <30 s: *p* < 0.0001, >120 s: *p* = 0.036; Figure 3A). This pattern was not observed in NREM sleep (Figure 3B). The change in REM sleep continuity was also reflected by a significant rise in the total number of brief awakenings (F (6, 84) = 3.584, *p* = 0.0033, two-way ANOVA with Šídák’s multiple comparisons test, <30 s: *p* < 0.0001; Figure 3C). Additionally, when examined by circadian phase, mClkΔ19 mice exhibited significantly more short REM sleep episodes (<30 s) and fewer long REM sleep episodes (>120 s) during the light phase compared to EYFP controls (F (6, 84) = 6.223, *p* < 0.0001, two-way ANOVA with Šídák’s multiple comparisons test, <30 s: *p* < 0.0001, <120 s: *p* = 0.007; Figure 3D), with no changes in wakefulness or NREM sleep (Figure 3E,F). During the dark phase, mClkΔ19 mice showed a notable increase in short REM sleep episodes (F (6, 84) = 2.171, *p* = 0.054; Šídák’s multiple comparisons test, <30 s: *p* = 0.001; Figure 3G), along with significantly more brief awakenings (F (6, 84) = 11.57, *p* < 0.0001, Šídák’s multiple comparisons test, <30 s: *p* < 0.0001; Figure 3I). There was also a trend toward longer NREM sleep episodes during the dark phase than in the EYFP mice (F (6, 84) = 1.804, *p* = 0.108, Šídák’s multiple comparisons test, <240 s: *p* = 0.056; Figure 3H). These findings indicate that suppression of CLOCK activity in VMH^PDYN+^ neurons primarily affects REM sleep stability during the light phase, resulting in fragmented REM sleep, and increases the frequency and instability of sleep–wake transitions during the dark phase.

To examine how suppression of CLOCK activity in VMH^PDYN+^ neurons influences sleep–wake transitions, we further analyzed the frequency of phase states. No significant differences were found over 24 h or during the light phase (Figure 4A–H). Interestingly, compared to EYFP mice, mClkΔ19 mice showed a significantly higher frequency of NREM and REM sleep episodes during the dark phase (REM: t (12) = 2.486, *p* = 0.029, NREM: t (12) = 2.381, *p* = 0.035, unpaired *t*-test; Figure 4I,J) and displayed a greater tendency toward wakefulness (t (12) = 2.171, *p* = 0.051, unpaired *t*-test; Figure 4K). Additionally, mClkΔ19 mice exhibited notably increased reciprocal transitions between wakefulness and NREM sleep compared to EYFP controls (F (3, 48) = 2.23, *p* = 0.097, two-way ANOVA with Šídák’s multiple comparisons test, W–N: *p* = 0.006, N–W: *p* = 0.03; Figure 4L). These results suggest that suppression of CLOCK activity in VMH^PDYN+^ neurons promotes irregular sleep–wake transitions, with partial modulation by circadian influences.

### 3.4. Functional Suppression of CLOCK Activity in VMH^PDYN+^ Neurons Prolongs Sleep Latency and Reduces REM Sleep Cycle During the Dark Phase

Since the suppression of CLOCK activity in VMH^PDYN+^ neurons primarily affected REM sleep, we further characterized REM sleep architecture by analyzing REM sleep cycles and latency. Compared with EYFP controls, mClkΔ19 mice exhibited a significantly longer latency to the first REM sleep episode (*p* = 0.029, U = 7, Mann–Whitney test; Figure 5A,B). In contrast, the duration of REM sleep cycles was significantly shorter in mClkΔ19 mice than in EYFP controls (*p* = 0.043, D = 0.708, Kolmogorov–Smirnov test; Figure 5C). Although REM cycle length did not differ between groups during the light phase (t (12) = 0.664, *p* = 0.519, unpaired *t*-test), mClkΔ19 mice displayed markedly shorter REM sleep cycles during the dark phase (t (12) = 3.101, *p* = 0.009, unpaired *t*-test; Figure 5D–F). Because REM sleep typically follows NREM sleep, we also examined the parameters of NREM sleep. Compared with EYFP controls, mClkΔ19 mice showed a significantly longer latency to the first NREM episode (*p* = 0.009, D = 0.833, Kolmogorov–Smirnov test; Figure 5G,H), whereas NREM cycle length remained unchanged (Figure 5I,L). Collectively, these findings indicate that suppression of CLOCK activity in VMH^PDYN+^ neurons alters REM sleep quantity and that the temporal organization of REM sleep becomes irregular in mice, suggesting a disruption in the coordinated regulation of REM timing and cycling.

### 3.5. Functional Suppression of CLOCK Activity in VMH^PDYN+^ Neurons Suppresses Gamma Oscillations During REM Sleep and Wakefulness

To further elucidate how the suppression of CLOCK activity in VMH^PDYN+^ neurons influences the quality and spectral characteristics of REM sleep, we analyzed EEG power across major frequency bands during wakefulness, NREM sleep, and REM sleep. In the light phase, mClkΔ19 mice showed a significant reduction in gamma energy during REM sleep compared with EYFP mice (F (3,48) = 7.658, *p* = 0.0003, two-way ANOVA with Šídák’s multiple comparisons test, Gamma: *p* = 0.0011; Figure 6A), whereas no significant differences were observed during NREM sleep or wakefulness (Figure 6B,C). In the dark phase, gamma energy in mClkΔ19 mice was significantly decreased during both REM sleep and wakefulness, while delta energy during wakefulness was markedly increased relative to controls; no significant changes were detected during NREM sleep (REM: F (3,48) = 5.472, *p* = 0.0026, Gamma: *p* = 0.0048; WAKE: F (3,48) = 7.144, *p* = 0.0005, Delta: *p* = 0.048, Gamma: *p* = 0.002; NREM: F (3,48) = 0.089, *p* = 0.966, two-way ANOVA with Šídák’s multiple comparisons test; Figure 6D–F). Together, these findings indicate that suppression of CLOCK activity in VMH^PDYN+^ neurons selectively alters EEG spectral composition across the sleep–wake cycle, particularly attenuating gamma oscillations during REM sleep and wakefulness.

## 4. Discussion

The present study demonstrates that the suppression of CLOCK activity in PDYN-expressing neurons of the VMH hinders REM sleep, reduces locomotor activity, and decreases theta-gamma oscillations during REM sleep and wakefulness in mice, which is dependent on the circadian rhythm. Previous studies have revealed that VMH-driven metabolic and aggressive regulation exhibits circadian rhythmicity or operates independently of the master clock, the SCN [10,15,33]. In vivo recordings revealed that 39.8% of the 78 recorded neurons in the VMH were associated with wakefulness, 33.3% of neurons were active during sleep phases, and 17 neurons exhibited increased activity during the dark phase, indicating diurnal rhythmic variations [20]. Meanwhile, lesions in the VMH of rats can prevent the increased locomotion trend during the light phase induced by the fasting model [34,35], and increase the duration of individual NREM and REM sleep episodes, as well as the frequency of REM sleep episodes during the dark phase [36]. However, whether the CLOCK activity in specific neurons of the VMH is involved in the sleep–wake cycle and locomotor activity related to circadian rhythm requires further exploration. Given the dense expression of PDYN neurons in the VMH and their role in the sleep–wake cycle [23,26,27], what is the functional role of the CLOCK activity in these specific neurons?

In this study, we generated neuron-type-specific ClockΔ19 mutant mice by targeting functional suppression of CLOCK activity in PDYN-expressing neurons within the VMH using AAV-mediated gene editing, and then subjected these mice to 24 h monitoring of sleep–wake cycles and locomotor activity. In the sleep–wake cycle, we found that suppression of CLOCK activity of PDYN neurons in the VMH markedly reduced the duration of REM sleep during the light phase (Figure 2), induced fragmentation of REM sleep, and increased short arousals (Figure 3), leading to an increase in NREM sleep to wakefulness transitions during the dark phase (Figure 4). Comparably, selectively removing circadian factor BMAL1 from histaminergic neurons in the tuberomammillary nucleus (TMN) induced more fragmented sleep, increased wakefulness during the light phase, and enhanced reciprocal transitions between NREM and REM sleep [37]. Furthermore, the circadian gene casein kinase 1 epsilon (Csnk1e) mutant mice also exhibited an increase in REM sleep during the dark period compared to wild-type controls [38]. Although *Clock* mutant mice show reduced NREM sleep and attenuated REM rebound [39], that model involves global, developmental replacement of endogenous CLOCK with mutant CLOCK. In contrast, our AAV-CLOCKΔ19 strategy produces an adult-onset, cell-type-specific functional suppression of CLOCK activity while endogenous wild-type CLOCK remains present. Our approach is not equivalent to eliminating *Clock* gene expression but instead achieves protein-level dominant-negative inhibition in PDYN neurons. Consequently, the sleep phenotypes observed in constitutive Clock mutant models should be interpreted as a conceptual reference rather than a direct comparator for the present manipulation, as differences in developmental timing, spatial specificity, and molecular mechanism are likely to yield distinct outcomes. Together, our findings suggest that the CLOCK activity in PDYN-expressing neurons of the VMH plays a critical role in circadian regulation of REM sleep by maintaining its duration and stability, and that its function may interact with other circadian genes in a circuit-specific manner. Given the neuronal heterogeneity within the VMH [40], whether disruption of the CLOCK activity in non-PDYN-expressing neurons similarly affects sleep–wake architecture? These questions require further investigation.

In the REM sleep cycle, we found that functional suppression of CLOCK activity of PDYN neurons in the VMH prolonged the latency of REM sleep and decreased the REM sleep cycles during the dark phase (Figure 5), indicating that maintaining a circadian rhythm is crucial for REM sleep homeostasis. Previous studies demonstrated that the circadian control of REM sleep is mediated by clock gene oscillations in the SCN [9,41,42]. Therefore, these findings raise the possibility that CLOCK activity in VMH-PDYN neurons may influence REM sleep through interactions with SCN circadian outputs or by functioning as an intrinsic regulatory node within the sleep–wake circuitry. CLOCKΔ19 is a dominant-negative mutant in which deletion of exon 19 abolishes CLOCK’s ability to transcriptionally activate E-box-dependent clock-controlled genes, while preserving heterodimerization with BMAL1. In homozygous mutant mice, this results in a lengthened circadian period that progressively transitions to arrhythmicity, highlighting the potency of CLOCKΔ19 as a functional inhibitor of circadian oscillator stability [43,44,45]. In the circadian index and locomotor activity, we further observed that the overexpression of CLOCK mutation in PDYN neurons of the VMH attenuated the circadian rhythm of the sleep–wake cycle, decreased free-running activity during the dark period, and induced an advanced activity peak (Figure 1). Consistently, deletion of the transcription factor Pou4f1 (*Brn3a*) in the dorsal medial habenula (dMHb) led to a decrease in voluntary wheel-running activity along with a more extended circadian period in mice [46]. Similarly, an advanced phase of locomotion has been reported in the period circadian regulator 2 (*Per2*) mutant mice [47]. In contrast, mClkΔ19 mice display a tendency toward an increased free-running period of their activity rhythm compared to controls [39], while removing *BMAL1* from histaminergic neurons does not affect circadian wheel-running behavior [37]. The inconsistency in results may stem from variations in experimental design regarding the duration of recording across different genotypes, as well as differences in operational methodologies. Meanwhile, tracing studies show the SCN sends direct projections to the VMH and heavily connects via the dorsomedial hypothalamus (DMH), and SCN disruption impairs VMH-regulated feeding behaviors [10,20], suggesting the SCN may rhythmically regulate VMH functions through these dual pathways. Consequently, a PDYN-neuron-specific CLOCK suppression in the VMH could potentially alter molecular clock dynamics and circadian pacemaker activity within the SCN through multisynaptic feedback mechanisms. However, it remains to be determined whether the discrepancies between global *Clock* mutations and cell-type-specific suppression arise from differential network compensations. Moreover, it is still unclear whether the observed alterations in sleep–wake architecture and locomotor activity following PDYN-neuron-specific CLOCK suppression depend on SCN rhythmic outputs. Finally, although we did not directly measure circadian transcriptional outputs (such as *Per* gene rhythms) in VMH^PDYN+^ neurons, the well-established dominant-negative mechanism of CLOCKΔ19 supports this interpretation. Future studies using cell-type-specific transcriptomics or reporter approaches will be necessary to directly evaluate molecular circadian disruption within these neurons and further confirm the functional impact of CLOCKΔ19 expression.

Delta activity reflects large-scale neuronal synchrony and cortical inhibition [48,49,50], its abnormal elevation during REM sleep and wakefulness thus points to a state of pathological hypoactivation [51,52,53]. The loss of gamma activity suggests impaired information processing and mnemonic function [54,55,56,57,58]. Notably, the gamma power showed a significant decrease in mClkΔ19 mice during REM sleep, a stage critical for emotional and mnemonic consolidation, which implies a breakdown of these integrative processes. Most strikingly, during the dark-phase wake period, when high arousal is expected, mClkΔ19 mice exhibited a “drowsy” EEG pattern characterized by elevated delta and diminished gamma power (Figure 6). This inversion of the normal spectral signature indicates a failure of the circadian arousal drive, leading to a persistently hypoaroused and cognitively constrained state. Therefore, in the follow-up study, the precise mechanisms by which CLOCK interference alters the EEG spectrum, especially reduced REM gamma and increased wake delta, remain speculative. We need to integrate fiber photometry or calcium imaging to directly monitor the activity and rhythmic changes in VMH^PDYN+^ neurons and investigate whether changes in the EEG frequency bands result in changes in cognitive levels by behavioral tests in mice.

Previous studies have found that the disruption of CLOCK can induce an elevated anxiety level and mania-like behavior [13,59]. REM sleep is known to regulate emotional behaviors [4,60,61,62]. Therefore, the observed reduction in REM sleep duration following *CLOCK* suppression in the VMH^PDYN+^ neurons may have significant implications for emotion-related behaviors in mice, a possibility that warrants systematic investigation. Given the established sexual dimorphism in VMH circuitry [63,64,65], our future study should specifically explore the potential gender differences associated with this circadian rhythm regulation, thereby further enhancing the generalizability of the conclusions. Additionally, we acknowledge that AAV delivery carries the possibility of some viral spread and off-target expression. In the description of the Methods and Figure 1, we emphasized the specificity of the VMH injection site and noted that despite potential off-target effects, the specific behavioral and physiological phenotypes we observed (e.g., reduced REM sleep and EEG changes) are highly consistent with literature reports of VMH dysfunction, strongly supporting the VMH^PDYN+^ circuit as the primary mediator.

## 5. Conclusions

In summary, our findings demonstrate that the CLOCK activity in VMH^PDYN+^ neurons is vital for synchronizing circadian rhythms with sleep–wake cycles. Suppression of CLOCK activity in these neurons leads to phase-dependent alterations in locomotor activity, REM sleep structure, and cortical oscillations, indicating that VMH^PDYN+^ neurons mediate connections between circadian and REM sleep processes. These findings expand our understanding of the hypothalamic mechanisms that control sleep and circadian rhythms, highlighting VMH^PDYN+^ circuits as promising targets for addressing sleep and rhythm disturbances.

## Figures and Tables

**Figure 1 neurolint-18-00005-f001:**
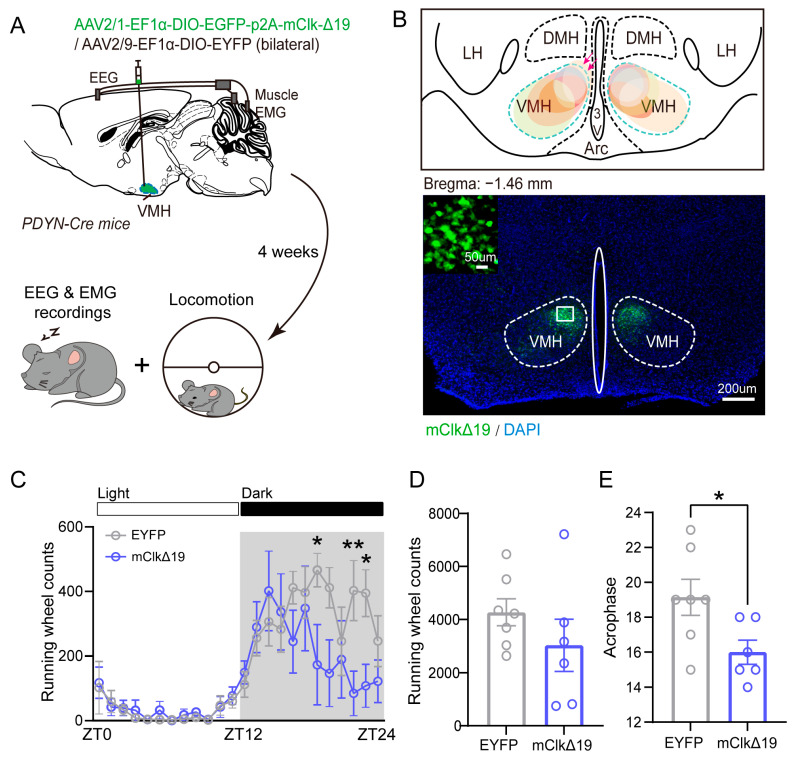
**Functional suppression of CLOCK activity in VMH^PDYN+^ neurons attenuates locomotion during the dark phase and affects the circadian rhythm of the sleep-wake cycle.** (**A**) Schematic of simultaneous EEG-EMG recordings and locomotor activity in the PDYN-Cre mice 4 weeks after virus (AAV2/1-EF1α-DIO-EGFP-p2A-mClk-Δ19 or AAV2/9-EF1α-DIO-EYFP) injection into the VMH. The experimental groups are divided into EYFP and mClkΔ19 mice. (**B**) Up: anatomical localization of viral injections in the VMH, outlined by the light blue dotted line. Color-coded regions represent the distribution of viral expression across all mice (*n* = 6). Pink arrows indicate occasional EGFP-positive cells outside the VMH. Bottom: A representative image showing virus expression in the bilateral VMH. Scale bar, 200 μm. The region enclosed by the square in the figure is shown in the upper-left corner of the image, highlighting cells labeled with green fluorescence. Scale bar, 50 μm. (**C**) The running wheel counts between EYFP and mClkΔ19 mice in 24 h. F (23, 264) = 1.873, *p* = 0.011, two-way ANOVA with Šídák’s multiple comparisons test, ZT19: *p* = 0.027; ZT20: *p* = 0.073; ZT22: *p* = 0.01; ZT23: *p* = 0.034. (**D**) Total running wheel counts between EYFP and mClkΔ19 mice in 24 h. (**E**) Time of peak exercise occurrence (acrophase) between EYFP and mClkΔ19 mice. *p* = 0.033, unpaired t-test, EYFP: 19.14 ± 1.033, mClkΔ19: 16 ± 0.683. (**F**) Total running wheel counts between EYFP and mClkΔ19 mice in the 12 h light phase. (**G**) Total running wheel counts between EYFP and mClkΔ19 mice in the 12 h dark phase. (**H**) Total running wheel counts between EYFP and mClkΔ19 mice in the post-6 h dark phase. *p* = 0.035, Mann–Whitney test. EYFP: 2168 ± 389.8, mClkΔ19: 824.4 ± 520.1. (**I**–**K**) The circadian index in REM sleep, NREM sleep, and wakefulness between EYFP and mClkΔ19 mice. REM: t (12) = 2.120, *p* = 0.056, unpaired *t*-test; NREM: t (12) = 2.296, *p* = 0.041, unpaired *t*-test; Wakefulness: t (12) = 2.447, *p* = 0.031, unpaired *t*-test. Asterisk, * *p* < 0.05, ** *p* < 0.01; *n* = 6–8. Group values were reported as mean ± standard error. ZT, Zeitgeber Time; VMH, the ventromedial hypothalamus.

**Figure 2 neurolint-18-00005-f002:**
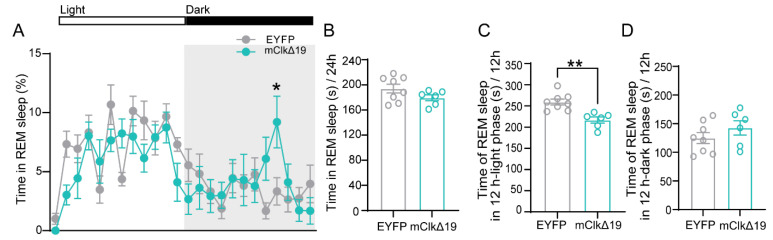
**Functional suppression of CLOCK activity in VMH^PDYN+^ neurons reduces the duration of REM sleep during the light phase.** (**A**) Time of day distribution of REM% in EYFP and mClkΔ19 mice while exposed to a 12 h light: 12 h dark cycle (LD 12:12 cycle). Two-way ANOVA test with Bonferroni’s multiple comparisons test, F (23, 288) = 1.912, *p* = 0.0074, ZT21: *p* = 0.041. (**B**) The average total time spent in REM sleep over 24 h between EYFP and mClkΔ19 mice. (**C**) The average time spent in REM sleep during the 12 h light phase for EYFP and mClkΔ19 mice. Unpaired *t*-test, t (12) = 4.156, *p* = 0.0013, EYFP: 259.1 ± 6.993 s, mClkΔ19: 216.6 ± 7.257 s. (**D**) The average time spent in REM sleep by mice during the 12 h dark phase for EYFP and mClkΔ19 mice. (**E**) Time of day distribution of NREM% in EYFP and mClkΔ19 mice while exposed to a 12 h light: 12 h dark cycle. (**F**) The average total time spent in NREM sleep over 24 h between EYFP and mClkΔ19 mice. (**G**) The average amount of time spent in REM sleep during the 12 h light phase for EYFP and mClkΔ19 mice. (**H**) The average time spent in REM sleep during the 12 h dark phase for EYFP and mClkΔ19 mice. (**I**) Time of day distribution of wakefulness % in EYFP and mClkΔ19 mice while exposed to a 12 h light: 12 h dark cycle. (**J**) The average total time spent in wakefulness over 24 h between EYFP and mClkΔ19 mice. (**K**) The average time spent in wakefulness during the 12 h light phase between EYFP and mClkΔ19 mice. (**L**) The average time spent awake during the 12 h dark phase in EYFP and mClkΔ19 mice. Asterisk, * *p* < 0.05, ** *p* < 0.01; *n* = 6–8. Group values were reported as mean ± standard error. ZT, Zeitgeber Time; VMH, the ventromedial hypothalamus.

**Figure 3 neurolint-18-00005-f003:**
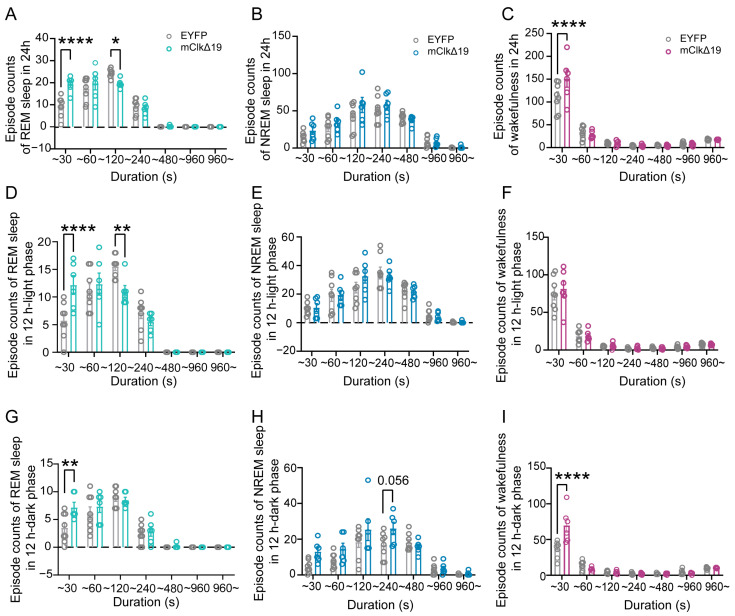
**Functional suppression of CLOCK activity in VMH^PDYN+^ neurons affects the sleep architecture.** (**A**–**C**) The total episode counts of REM sleep, NREM sleep, and wakefulness over 24 h between EYFP and mClkΔ19 mice. REM: two-way ANOVA with Sidak’s multiple comparisons test, F (6, 84) = 7.271, *p* < 0.0001, ~30 s: *p* < 0.0001, ~120 s: *p* = 0.036; Wakefulness: F (6, 84) = 3.584, *p* = 0.0033, ~30 s: *p* < 0.0001. (**D**–**F**) The total episode counts of REM sleep, NREM sleep, and wakefulness in the 12 h light phase between EYFP and mClkΔ19 mice. REM: two-way ANOVA with Sidak’s multiple comparisons test, F (6, 84) = 6.223, *p* < 0.0001, ~30 s: *p* < 0.0001, ~120 s: *p* = 0.007. (**G**–**I**) The total episode counts of REM sleep, NREM sleep, and wakefulness in the 12 h dark phase between EYFP and mClkΔ19 mice. REM: two-way ANOVA with Sidak’s multiple comparisons test, F (6, 84) = 2.171, *p* = 0.054, ~30 s: *p* = 0.001. NREM: F (6, 84) = 1.804, *p* = 0.108, ~30 s: *p* = 0.056. Wakefulness: F (6, 84) = 11.57, *p* < 0.0001, ~30 s: *p* < 0.0001. Asterisk, * *p* < 0.05, ** *p* < 0.01, **** *p* < 0.0001; *n* = 6–8. Group values were reported as mean ± standard error. VMH, the ventromedial hypothalamus.

**Figure 4 neurolint-18-00005-f004:**
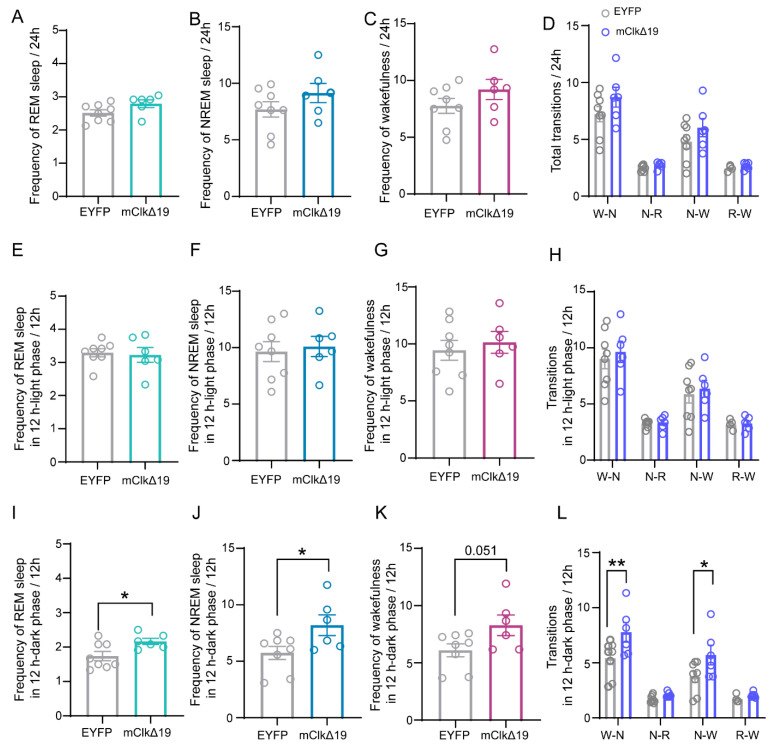
**Functional suppression of CLOCK activity in VMH^PDYN+^ neurons enhances sleep frequency and sleep–wake transitions during the dark phase**. (**A**–**C**) The mean frequencies of REM sleep, NREM sleep, and wakefulness over 24 h in EYFP and mClkΔ19 mice. (**D**) The average frequency of transitions within 24 h between EYFP and mClkΔ19 mice. (**E**–**G**) The mean frequency of REM sleep, NREM sleep, and wakefulness in the 12 h light phase. (**H**) The mean frequency of transitions in the 12 h light phase between EYFP and mClkΔ19 mice. (**I**–**K**) The mean frequency of REM sleep, NREM sleep, and wakefulness in the 12 h dark phase between EYFP and mClkΔ19 mice. Unpaired *t*-test, REM: *p* = 0.029, t (12) = 2.486; NREM: *p* = 0.035, t (12) = 2.381; Wakefulness: *p* = 0.051; t (12) = 2.171. (**L**) The mean frequency of transitions in the 12 h dark phase between EYFP and mClkΔ19 mice. Two-way ANOVA test with Sidak’s multiple comparisons test, F (3, 48) = 2.23, *p* = 0.0967, W–N: *p* = 0.006; N–W: *p* = 0.03. Asterisk, * *p* < 0.05, ** *p* < 0.01; *n* = 6–8. Group values were reported as mean ± standard error. VMH, the ventromedial hypothalamus.

**Figure 5 neurolint-18-00005-f005:**
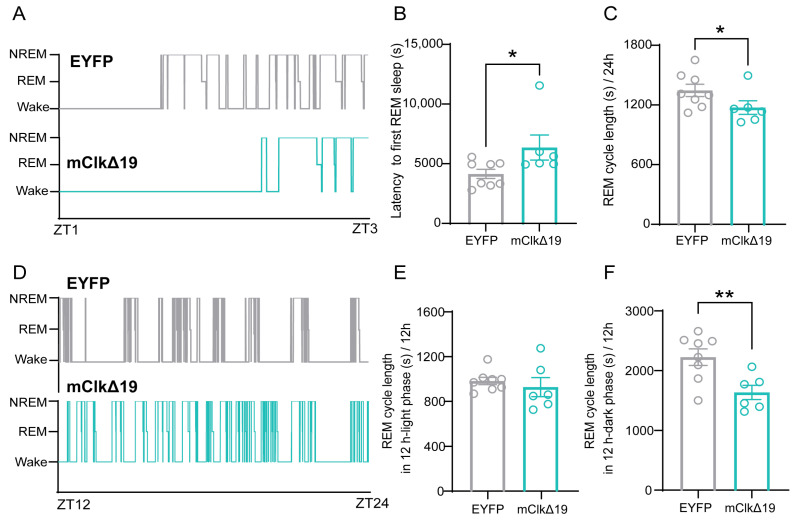
**Functional suppression of CLOCK activity in VMH^PDYN+^ neurons prolongs sleep latency and reduces REM sleep cycle during the dark phase.** (**A**) The representative diagram of the sleep architecture of EYFP and mClkΔ19 mice during the first 2 h of the light phase, including the initial REM sleep occurrence. The top black line indicates the EYFP group, and the bottom green line represents the mClkΔ19 group. (**B**) The latency from NREM to first REM sleep between EYFP and mClkΔ19 mice. Mann–Whitney test, *p* = 0.029. EYFP: 4149 ± 382.4 s, mClkΔ19: 6359 ± 1052 s. (**C**) The average REM cycle length over 24 h between EYFP and mClkΔ19 mice. Kolmogorov–Smirnov test, *p* = 0.043, EYFP: 1342 ± 61.91s, mClkΔ19: 1170 ± 68.37 s. (**D**) The representative diagram of the sleep architecture of mClkΔ19 mice during the 12 h dark phase, which aligns with figure F. The top black line indicates the EYFP group, and the bottom green line represents the mClkΔ19 group. (**E**) The mean REM cycle length in the 12 h light phase between EYFP and mClkΔ19 mice. Unpaired *t*-test, t (12) = 0.664, *p* = 0.519, EYFP: 957.7 ± 32.23 s, mClkΔ19: 904.1 ± 83.73 s. (**F**) The mean REM cycle length in the 12 h dark phase between EYFP and mClkΔ19 mice. Unpaired *t*-test, t (12) = 3.101, *p* = 0.009. EYFP: 2194 ± 136.7 s, mClkΔ19: 1609 ± 118 s. (**G**) The representative diagram of the sleep architecture of EYFP (top) and mClkΔ19 mice (bottom) during the first 2 h of the light phase, including the initial NREM sleep occurrence. (**H**) The latency to first NREM sleep between EYFP and mClkΔ19 mice. Kolmogorov–Smirnov test, *p* = 0.009, EYFP: 1676 ± 358.3 s, mClkΔ19: 4536 ± 1261 s. (**I**) The average NREM cycle length over 24 h between EYFP and mClkΔ19 mice. Kolmogorov–Smirnov test, *p* = 0.620, EYFP: 480.7 ± 54.29 s, mClkΔ19: 403.4 ± 36.53 s. (**J**) The representative diagram of the sleep architecture of EYFP (top) and mClkΔ19 mice (bottom) during the 12 h dark phase, which corresponds to figure L. (**K**) The mean NREM cycle length in the 12 h light phase between EYFP and mClkΔ19 mice. Unpaired *t*-test, t (12) = 0.664, *p* = 0.565, EYFP: 374.4 ± 37.43 s, mClkΔ19: 346.1 ± 28.40 s. (**L**) The mean NREM cycle length in the 12 h dark phase between EYFP and mClkΔ19 mice. Kolmogorov–Smirnov test, *p* = 0.302, EYFP: 673.7± 96.14 s, mClkΔ19: 497.7 ± 66.64 s. Asterisk, * *p* < 0.05, ** *p* < 0.01; *n* = 6–8. Group values were reported as mean ± standard error.

**Figure 6 neurolint-18-00005-f006:**
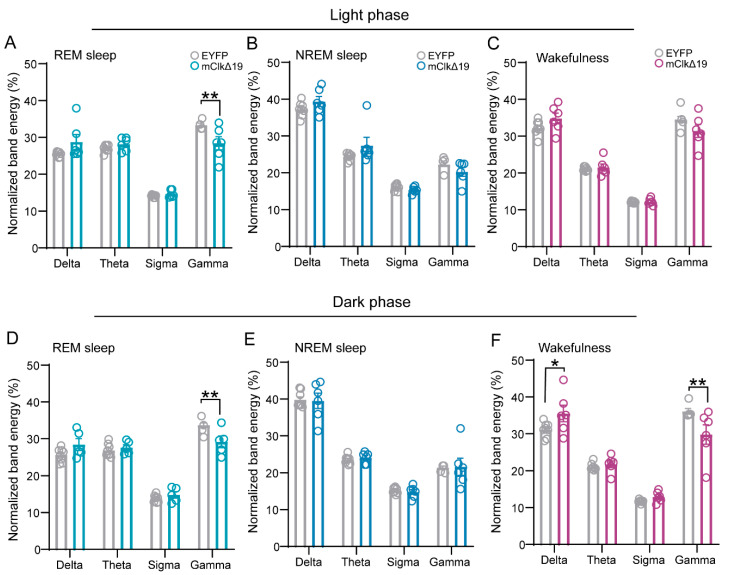
Functional suppression of CLOCK activity in VMH^PDYN+^ neurons suppresses gamma oscillations during REM sleep and wakefulness. (**A**) Normalized band energy (%) during REM sleep in the light phase. F (3, 48) = 7.658, *p* = 0.0003, two-way ANOVA with Šídák’s multiple comparisons test, Gamma: *p* = 0.0011, Delta: *p* = 0.053, Theta: *p* = 0.817, Sigma: *p* = 0.991. (**B**) Normalized band energy (%) during NREM sleep in the light phase. F (3, 48) = 2.953, *p* = 0.042, two-way ANOVA with Šídák’s multiple comparisons test, Gamma: *p* = 0.486, Delta: *p* = 0.436, Theta: *p* = 0.121, Sigma: *p* = 0.973. (**C**) Normalized band energy (%) during wakefulness in the light phase. F (3, 48) = 3.596, *p* = 0.020, two-way ANOVA with Šídák’s multiple comparisons test, Gamma: *p* = 0.059, Delta: *p* = 0.163, Theta: *p* = 0.994, Sigma: *p* > 0.999. (**D**) Normalized band energy (%) during REM sleep in the dark phase. F (3, 48) = 5.472, *p* = 0.0026, two-way ANOVA with Šídák’s multiple comparisons test, Gamma: *p* = 0.0048, Delta: *p* = 0.283, Theta: *p* = 0.907, Sigma: *p* = 0.853. (**E**) Normalized band energy (%) during NREM sleep in the dark phase. F (3, 48) = 0.089, *p* = 0.966, two-way ANOVA with Šídák’s multiple comparisons test, Gamma: *p* > 0.999, Delta: *p* = 0.999, Theta: *p* = 0.994, Sigma: *p* = 0.998. (**F**) Normalized band energy (%) during wakefulness in the dark phase. F (3, 48) = 7.144, *p* = 0.0005, two-way ANOVA with Šídák’s multiple comparisons test, Gamma: *p* = 0.002, Delta: *p* = 0.048, Theta: *p* = 0.990, Sigma: *p* = 0.924. Asterisk, * *p* < 0.05, ** *p* < 0.01, *n* = 6–8. Group values were reported as mean ± standard error. VMH, the ventromedial hypothalamus.

## Data Availability

All raw data from the study are available by contacting the corresponding author.

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
