# Peer review of "Functional Suppression of CLOCK Activity in Ventromedial Hypothalamic Prodynorphin Neurons Alters Locomotor Activity and Rapid Eye Movement Sleep"

_2035-8377, 2025, doi:10.3390/neurolint18010005_

Round 1

Reviewer 1 Report

Comments and Suggestions for Authors

This study aims to test the importance of protodynorphin expressing neurons in the VMH to circadian and sleep homeostasis.  The manuscript is well written, the figures are clearly presented, and statistical validation of their results was sound.  Futhermore, the sleep measures employed are nuanced, thorough, and well explained. However, this reviewer was confused by the explanation of the approach employed - specifically with AAV overexpression of delta19Clock.

The authors us Cre-expressing PDYN mice to drive delta19 CLOCK mt expression in neurons expressing Cre.  However:

  • More details about the vector design would be helpful.  No mention of whether mouse or human sequence was used for vector design - although "mCLOCK" used suggests mouse.  A figure with the vector map, etc. would be helpful as well as more details about design.
  • Why delta19 was chosen was not clearly explained other than “disruption of the Clock gene has been shown to impair sleep architecture and circadian rhythms..”  However, the authors are not impacting Clock gene expression directly, they are targeting CLOCK protein function by overexpressing a dominant negative.  This issue is evident throughout the manuscript.
  • Delta19 causes a long period phenotype that degrades into arythmicity in mammals but this is not mentioned nor is this information used to help explain any aspect of their findings.
  • Overexpression of delta19 means the endogenous CLOCK allele is still present.  Therefore, a direct measure of circadian dyshomeostasis at the molecular level would be helpful in supporting the authors’ conclusions that their AAV construct functioned appropriately.
  • “Disruption of the Clock gene in VMHPDYN+ neurons attenuates dark-phase locomotion and affects the circadian rhythm of the sleep-wake cycle”
    • Authors are not impacting gene expression, they are impacting protein function.  This should be corrected throughout the manuscript
  • “To better understand how the Clock mutation in VMHPDYN+ neurons affects REM sleep architecture,” seems misleading.  The authors are overexpressing the CLOCKmt not mutating the Clock allele in PDYN neuron.  Thus, the endogenous mouse Clock allele is still intact.
  • “Unexpectedly, the duration of NREM sleep in Clock mutant mice was one to two hours shorter and showed smaller increases in REM sleep compared to the controls [39].”
    • This comparison is somewhat invalid as the model they are comparing theirs to expresses the mtCLOCK in place of WT, where as the authors overexpress mtCLOCK and wt mouse CLOCK is still present.  This should be discussed.
    • In figure 1, there appears to be GFP in cells outside of the VMH.  This is not mentioned.

Overall, a better justification and explaination for using the AAV-delta19 Clock approach is needed.  Data interpretation and conclusions should be modified to reflect this.

Author Response

Dear reviewer,

Thank you for your comments and professional advice. These opinions help to improve the academic rigor of our manuscript. Based on your suggestion and request, we have made the corrected modifications to the revised manuscript. We hope that our work can be improved again. Please see the attachment,Furthermore, we would like to show the details as follows:

Responses to comments from Reviewer #1

Q1: More details about the vector design would be helpful. No mention of whether mouse or human sequence was used for vector design - although "mCLOCK" used suggests mouse. A figure with the vector map, etc. would be helpful as well as more details about design.

A: Thank you for this helpful comment. We have revised the manuscript to provide more complete information about the viral construct. The AAV2/1-Ef1α-DIO-EGFP-p2A-mClkΔ19 plasmid used in this study was obtained as a gift from the Erquan Zhang Laboratory (National Institute of Biological Sciences, Beijing), and therefore, the original cloning and sequence design were not performed in our laboratory. Accordingly, the exact nucleotide sequence is not available to us.

However, we have now added all known design information: the vector contains an Ef1α promoter driving a Cre-dependent DIO cassette, followed by EGFP, a self-cleaving p2A sequence, and the mouse CLOCKΔ19 (mClkΔ19) dominant-negative coding sequence. To further improve clarity, we have included a schematic vector map illustrating the arrangement of these functional elements. This schematic is intended to aid readers but does not represent nucleotide-level detail, which we do not possess.

The vector schematic has been added as Supplementary Figure S1, and the Methods section has been updated accordingly. In the “Methods” of the manuscript (Page 3, lines 8-13): “The AAV2/1-Ef1α-DIO-EGFP-p2A-mClkΔ19 construct was a generous gift from the Erquan Zhang Laboratory (NIBS, Beijing). Because the plasmid was not designed or cloned in our laboratory, the exact nucleotide sequence is not available. The cassette expresses the mouse CLOCKΔ19 dominant-negative protein and includes an Ef1α promoter, a Cre-dependent DIO inversion system, EGFP, a p2A self-cleaving peptide, and the mClkΔ19 coding sequence. A schematic illustration of the vector design is provided in Supplementary Figure S1 (Page 27).”

Supplementary Figure S1: Schematic map of the AAV2/1-EF1α-DIO-EGFP-p2A-mClkΔ19 viral construct used in this study.

This diagram illustrates the functional organization of the viral vector, which was generously provided by the Erquan Zhang Laboratory (NIBS, Beijing). The construct contains an EF1α promoter driving a Cre-dependent double-floxed inverted open reading frame (DIO) to restrict expression to PDYN-Cre–positive neurons. The expression cassette includes EGFP followed by a self-cleaving p2A peptide and the mouse CLOCKΔ19 (mClkΔ19) dominant-negative coding sequence. The CLOCKΔ19 variant lacks exon 19 and functions by competitively inhibiting endogenous CLOCK-BMAL1 transcriptional activity. WPRE and polyA elements are included to enhance transcript stability and expression efficiency. Because the plasmid was obtained as a gift and not designed by our laboratory, the schematic reflects functional elements rather than nucleotide-level sequence information.”

Q2: Why delta19 was chosen was not clearly explained other than “disruption of the Clock gene has been shown to impair sleep architecture and circadian rhythms.” However, the authors are not impacting Clock gene expression directly, they are targeting CLOCK protein function by overexpressing a dominant negative. This issue is evident throughout the manuscript.

A: We thank the reviewer for this important point. We have now clarified the rationale and mechanism of using CLOCKΔ19 in the revised manuscript. CLOCKΔ19 is a well-characterized dominant-negative mutant in which deletion of exon 19 abolishes the ability of CLOCK to transcriptionally activate E-box–dependent clock-controlled genes while preserving its ability to heterodimerize with BMAL1. As a result, CLOCKΔ19 competes with endogenous wild-type CLOCK for BMAL1 binding and thereby suppresses CLOCK–BMAL1 transcriptional activity. Thus, our approach does not eliminate Clock gene expression; instead, it induces cell-type-specific functional inhibition of CLOCK at the protein level within PDYN neurons.

We chose CLOCKΔ19 because (1) it enables temporally precise, adult-onset suppression without developmental compensation present in germline Clock mutants, and (2) it provides a widely validated method to suppress local circadian transcriptional output in a cell-restricted manner. We have revised the Introduction and Methods to explicitly explain this rationale and replaced all wording that inaccurately implied gene deletion with terminology such as “AAV-mediated overexpression of dominant-negative CLOCKΔ19” or “functional suppression of CLOCK activity.”

As shown in the “Introduction” of the manuscript (Page 2, paragraph 3, lines 3-11): “…thereby overexpressing dominant-negative CLOCKΔ19 in PDYN neurons. CLOCKΔ19 is a dominant-negative variant of the CLOCK protein in which deletion of exon 19 eliminates its transcriptional activation function while leaving BMAL1 dimerization intact. When overexpressed, CLOCKΔ19 competitively binds BMAL1 and suppresses endogenous CLOCK-BMAL1-driven transcriptional oscillations. Importantly, this manipulation does not delete or silence the endogenous Clock gene; instead, it produces cell-type-specific functional inhibition of the CLOCK protein, allowing us to evaluate the local contribution of CLOCK signaling within PDYN neurons without developmental confounds associated with germline Clock mutations.”

As shown in the “Methods” of the manuscript (Page 3, lines 1-10): “The AAV2/1-Ef1α-DIO-EGFP-p2A-mClkΔ19 construct was a generous gift from the Erquan Zhang Laboratory (NIBS, Beijing). Because the plasmid was not designed or cloned in our laboratory, the exact nucleotide sequence is not available. The cassette expresses the mouse CLOCKΔ19 dominant-negative protein and includes an Ef1α promoter, a Cre-dependent DIO inversion system, EGFP, a p2A self-cleaving peptide, and the mClkΔ19 coding sequence. A schematic illustration of the vector design is provided in Supplementary Figure S1. The mClkΔ19 sequence used in this study encodes the mouse CLOCKΔ19 dominant-negative protein, which lacks exon 19 and competitively inhibits CLOCK-BMAL1 transcriptional activity. Thus, AAV-mediated expression of mClkΔ19 functionally suppresses CLOCK activity without altering endogenous Clock gene expression.”

 Q3: Delta19 causes a long period phenotype that degrades into arythmicity in mammals but this is not mentioned nor is this information used to help explain any aspect of their findings.

A: We agree that the well-characterized circadian phenotype of CLOCKΔ19, which initially lengthens the free-running period and eventually degrades into arrhythmicity in mammals, was not explicitly mentioned in the original manuscript. In our study, AAV-mediated overexpression of CLOCKΔ19 in PDYN neurons similarly suppresses local CLOCK–BMAL1 activity, providing a mechanistic basis for the observed alterations in REM sleep architecture and sleep fragmentation. We have added these points to the Discussion (Page 9, paragraph 2, lines 8-12) to clarify how the known long-period and arrhythmic phenotype of CLOCKΔ19 supports our interpretation of the sleep and locomotor phenotypes observed in the PDYN neuron-specific manipulation.

As shown in the “Discussion” of the manuscript (Page 9, paragraph 2, lines 8-12):

“CLOCKΔ19 is a dominant-negative mutant in which deletion of exon 19 abolishes CLOCK’s ability to transcriptionally activate E-box–dependent clock-controlled genes, while preserving heterodimerization with BMAL1. In homozygous mutant mice, this results in a lengthened circadian period that progressively transitions to arrhythmicity, highlighting the potency of CLOCKΔ19 as a functional inhibitor of circadian oscillator stability [43-45].”

References

  • Vitaterna MH et al. Mutagenesis and mapping of a mouse gene, Clock, essential for circadian behavior. Science 1994.
  • King DP et al. Positional cloning of the mouse circadian Clock gene. Cell 1997.
  • Debruyne, Jason P et al. A clock shock: mouse CLOCK is not required for circadian oscillator function. Neuron 2006.

Q4: Overexpression of delta19 means the endogenous CLOCK allele is still present.  Therefore, a direct measure of circadian dyshomeostasis at the molecular level would be helpful in supporting the authors’ conclusions that their AAV construct functioned appropriately.

A: We appreciate the reviewer’s insightful comment. We fully agree that AAV-mediated expression of CLOCKΔ19 does not eliminate endogenous Clock, and therefore direct molecular evidence of circadian disruption would strengthen the interpretation of our functional data.

At present, however, we were unable to perform molecular readouts such as rhythmic expression of E-box–dependent clock-controlled genes (e.g., Per1, Per2) specifically in VMHPDYN+ neurons due to technical limitations related to cell-type specificity and tissue amount. We acknowledge this as a limitation and now explicitly state it in the revised manuscript. Importantly, CLOCKΔ19 is a widely validated dominant-negative mutant whose mechanism of action is well established: deletion of exon 19 abolishes transcriptional activation but preserves BMAL1 heterodimerization, allowing CLOCKΔ19 to competitively inhibit endogenous CLOCK–BMAL1 activity. This mechanism has been demonstrated in multiple prior studies and underpins our use of this construct.

To address the reviewer’s point, we have added statements in both the Results and Discussion clarifying that: Our manipulation induces functional suppression, not deletion, of CLOCK activity. Molecular confirmation of circadian transcriptional dysregulation in PDYN neurons would further strengthen our conclusions, but such measurements are beyond the scope of the current study and will be pursued in future work. The behavioral and sleep phenotypes we observe are consistent with reduced local CLOCK–BMAL1 output, supporting the notion that the dominant-negative construct functioned as expected.

We have added the following text to the Discussion (Page 10, lines 2-8):

“Finally, although we did not directly measure circadian transcriptional outputs (such as Per gene rhythms) in VMHPDYN+ neurons, the well-established dominant-negative mechanism of CLOCKΔ19 supports this interpretation. Future studies using cell-type-specific transcriptomics or reporter approaches will be necessary to directly evaluate molecular circadian disruption within these neurons and further confirm the functional impact of CLOCKΔ19 expression.”

Q5: “Disruption of the Clock gene in VMHPDYN+ neurons attenuates dark-phase locomotion and affects the circadian rhythm of the sleep-wake cycle”. Authors are not impacting gene expression; they are impacting protein function. This should be corrected throughout the manuscript.

A: We thank the reviewer for this important clarification. We agree that our AAV–CLOCKΔ19 manipulation does not disrupt Clock gene expression but instead produces a dominant-negative inhibition of CLOCK protein function in PDYN-expressing VMH neurons. In the revised manuscript, we have corrected all wording that previously implied genetic deletion or gene disruption. These terms have been replaced with more accurate descriptions, including:

  • AAV-mediated overexpression of dominant-negative CLOCKΔ19
  • Functional suppression of CLOCK activity
  • Protein-level inhibition of CLOCK–BMAL1 transcriptional function

All relevant statements in the Abstract, Introduction, Results, Discussion, and Figure Legends have been updated accordingly to ensure mechanistic accuracy.

 In the “Results” section of the manuscript (Pages 5-8) and Figures and Figure legends” section of the manuscript (Pages 18-26): “Functional suppression of CLOCK activity in VMHPDYN+ neurons attenuates dark-phase locomotion and affects the circadian rhythm of the sleep-wake cycle.

Q6: “To better understand how the Clock mutation in VMHPDYN+ neurons affects REM sleep architecture,” seems misleading.  The authors are overexpressing the CLOCKmt not mutating the Clock allele in PDYN neuron. Thus, the endogenous mouse Clock allele is still intact.

A: We thank the reviewer for this important clarification. We agree that the original phrasing, “how the Clock mutation in VMHPDYN+ neurons affects REM sleep architecture”, is misleading. Our experimental design does not involve mutating the endogenous Clock allele in PDYN neurons. Instead, we express the dominant-negative CLOCKΔ19 protein via AAV, while the endogenous wild-type Clock allele remains intact and continues to produce functional CLOCK protein.

To address this concern, we have revised the manuscript throughout. All instances of “Clock mutation” have been replaced with terminology that accurately reflects the mechanism of our manipulation, such as:

“AAV-mediated overexpression of dominant-negative CLOCKΔ19,” “Functional inhibition of CLOCK activity,” or “dominant-negative suppression of CLOCK function in VMHPDYN+ neurons.”

We have revised the corresponding sentence in the Results section (Page 6, lines 25) to:

“To better understand how dominant-negative suppression of CLOCK activity in VMHPDYN+ neurons affects REM sleep architecture, …”

This modification clarifies that our study investigates the functional consequence of expressing the CLOCKΔ19 dominant-negative protein rather than altering the endogenous gene.

Q7: “Unexpectedly, the duration of NREM sleep in Clock mutant mice was one to two hours shorter and showed smaller increases in REM sleep compared to the controls [39].” This comparison is somewhat invalid as the model they are comparing theirs to expresses the mtCLOCK in place of WT, where as the authors overexpress mtCLOCK and wt mouse CLOCK is still present. This should be discussed.

A: We thank the reviewer for this insightful comment. We agree that the comparison to the study reporting reduced NREMS and attenuated REMS rebound in Clock mutant mice (Ref. 39) requires clarification. In that model, the endogenous Clock allele is replaced by a mutant form (mtCLOCK), resulting in a global and constitutive alteration of CLOCK function throughout development. By contrast, our approach involves AAV-mediated overexpression of dominant-negative CLOCKΔ19 in adult PDYN neurons, while endogenous wild-type CLOCK remains present. Thus, the molecular perturbation in our study is cell-type specific, adult-onset, and partial, rather than whole-animal and developmental.

We have now revised the Discussion to explicitly acknowledge that these mechanistic differences limit the direct comparability between the two models. Instead of implying equivalence, we now emphasize that the prior study provides a useful conceptual reference for understanding how CLOCK dysfunction can influence sleep architecture, but its phenotype should not be interpreted as a direct prediction for the outcome of our cell-restricted CLOCKΔ19 manipulation. The relevant sentences have been rewritten accordingly in the revised manuscript.

In the “Discussion” section of the manuscript (Pages 8-9, paragraph 3, lines 13-22): “Although Clock mutant mice show reduced NREM sleep and attenuated REM rebound [39] , that model involves global, developmental replacement of endogenous CLOCK with mutant CLOCK. In contrast, our AAV–CLOCKΔ19 strategy produces an adult-onset, cell-type–specific functional suppression of CLOCK activity while endogenous wild-type CLOCK remains present. We have now clarified this rationale and the mechanistic distinction in the revised manuscript, as our approach is not equivalent to eliminating Clock gene expression but instead achieves protein-level dominant-negative inhibition in PDYN neurons. Consequently, the sleep phenotypes observed in constitutive Clock mutant models should be interpreted as a conceptual reference rather than a direct comparator for the present manipulation, as differences in developmental timing, spatial specificity, and molecular mechanism are likely to yield distinct outcomes.”

Q8: In figure 1, there appears to be GFP in cells outside of the VMH. This is not mentioned.

A: We thank the reviewer for pointing out the presence of GFP signal outside the VMH in Figure 1B. We have carefully examined our data and acknowledge that some off-target expression occurs in neighboring regions, likely due to viral spread beyond the intended injection site. Although most expression is confined to the VMH, we now explicitly note this in the revised Results section, Methods section and have added arrows/annotations in Figure 1 to indicate off-target cells. Significantly, the behaviors we report are predominantly associated with VMHPDYN+ neurons, and our negative control (EYFP-only group) helps account for potential off-target effects.

In the “Results” section of the manuscript (Page 5, paragraph 3, lines 7-11): “Although viral expression was primarily restricted to the VMH, a small number of EGFP-positive cells were occasionally observed in adjacent hypothalamic regions (Figure 1B, pink arrows). These off-target cells were sparse and did not appear to influence the behavioral results, as confirmed by the EYFP-only control group.”

In the “Methods” section of the manuscript (Page 3, paragraph 2, lines 19-22): “Viral injections were targeted to the VMH; however, occasional spread to nearby hypothalamic regions was observed. Analysis focused on VMH-localized neurons, and EGFP-only controls were included to account for any off-target expression.”

In the “Figure 1 and Figure legends” section of the manuscript (Pages 18-19, lines 1–9):

Figure 1: Functional suppression of CLOCK activity in VMHPDYN+ neurons attenuates locomotion during the dark phase and affects the circadian rhythm of the sleep-wake cycle.

(A) Schematic of simultaneous EEG-EMG recordings and locomotor activity in the PDYN-Cre mice 4 weeks after virus (AAV2/1-EF1α-DIO-EGFP-p2A-mClk-Δ19 or AAV2/9-EF1α-DIO-EYFP) injection into the VMH. The experimental groups are divided into EYFP and mClkΔ19 mice.

(B) Up: anatomical localization of viral injections in the VMH, outlined by the light blue dotted line. Color-coded regions represent the distribution of viral expression across all mice (n = 6). Pink arrows indicate occasional EGFP-positive cells outside the VMH…”

Q9: Overall, a better justification and explaination for using the AAV-delta19 Clock approach is needed. Data interpretation and conclusions should be modified to reflect this.

A: We thank the reviewer for this important point. In the revised manuscript, we now provide a clearer and more detailed justification for employing the AAV–CLOCKΔ19 approach and have modified the interpretation of our results accordingly.

CLOCKΔ19 is a well-characterized dominant-negative allele in which deletion of exon 19 abolishes the transcriptional activation function of CLOCK on E-box–dependent target genes, while leaving its ability to heterodimerize with BMAL1 intact. As a result, CLOCKΔ19 effectively competes with endogenous CLOCK for BMAL1 binding and suppresses CLOCK–BMAL1–mediated transcriptional output. Importantly, this strategy does not delete the Clock gene; instead, it provides a cell-type-specific functional inhibition at the protein level within PDYN neurons.

We chose this approach because:

  • AAV-mediated expression enables temporally precise, adult-onset suppression of CLOCK function, avoiding developmental compensation present in germline Clock mutants.
  • CLOCKΔ19 has been widely validated as a robust tool for reducing circadian transcriptional output in a spatially and cell-restricted manner. Furthermore, CLOCKΔ19’s well-documented behavioral phenotype, characterized by a lengthened circadian period that ultimately leads to arrhythmicity, underscores its potency as a functional inhibitor of oscillator stability. This provides a mechanistic justification for using it to probe whether dampening local CLOCK-dependent transcription alters PDYN neuronal activity or behavioral responses.

In the revised Introduction (Page 2, paragraph 3, lines 3-11) and Methods (Page 3, lines 8-17), we now explicitly state these points and have replaced all inaccurate wording implying gene deletion with more precise terminology such as “AAV-mediated overexpression of dominant-negative CLOCKΔ19” or “functional suppression of CLOCK activity.” Corresponding adjustments to the Results (Page 5, paragraph 3, lines 1-11) and Discussion (Page 9, paragraph 2, lines 1-8) were made to ensure our conclusions accurately reflect the mechanism of action of CLOCKΔ19.

 In the revised Introduction of the manuscript (Page 2, paragraph 3, lines 3-20): “…overexpressing dominant-negative CLOCKΔ19 in PDYN neurons. CLOCKΔ19 is a dominant-negative variant of the CLOCK protein in which deletion of exon 19 eliminates its transcriptional activation function while leaving BMAL1 dimerization intact. When overexpressed, CLOCKΔ19 competitively binds BMAL1 and suppresses endogenous CLOCK–BMAL1–driven transcriptional oscillations. Importantly, this manipulation does not delete or silence the endogenous Clock gene; instead, it produces cell-type-specific functional inhibition of the CLOCK protein, allowing us to evaluate the local contribution of CLOCK signaling within PDYN neurons without developmental confounds associated with germline Clock mutations. Through 24-hour polysomnographic recordings and wheel-running activity monitoring, we found that functional suppression of CLOCK activity in VMHPDYN⁺ neurons attenuated nighttime locomotor activity and altered the circadian distribution of sleep-wake states. This manipulation selectively reduced REM sleep duration, increased REM fragmentation, disturbed sleep-wake transitions and REM cycling during the dark phase. Moreover, EEG spectral analysis revealed decreased gamma activity during REM sleep in the light phase and increased delta activity coupled with decreased gamma during wakefulness in the dark phase. Together, these findings demonstrate that the CLOCK activity in VMHPDYN⁺ neurons is crucial for maintaining REM sleep quality and coordinating sleep-wake rhythms within the circadian cycle.”

 In the Methods of the manuscript (Page 3, lines 8-17): “The AAV2/1-Ef1α-DIO-EGFP-p2A-mClkΔ19 construct was a generous gift from the Erquan Zhang Laboratory (NIBS, Beijing). Because the plasmid was not designed or cloned in our laboratory, the exact nucleotide sequence is not available. The cassette expresses the mouse CLOCKΔ19 dominant-negative protein and includes an Ef1α promoter, a Cre-dependent DIO inversion system, EGFP, a p2A self-cleaving peptide, and the mClkΔ19 coding sequence. A schematic illustration of the vector design is provided in Supplementary Figure S1. The mClkΔ19 sequence used in this study encodes the mouse CLOCKΔ19 dominant-negative protein, which lacks exon 19 and competitively inhibits CLOCK-BMAL1 transcriptional activity. Thus, AAV-mediated expression of mClkΔ19 functionally suppresses CLOCK activity without altering endogenous Clock gene expression.”

 In the Results of the manuscript (Page 5, paragraph 3, lines 1-11): “3.1 Functional suppression of CLOCK activity in VMHPDYN+ neurons attenuates dark-phase locomotion and affects the circadian rhythm of the sleep-wake cycle

To examine the role of circadian rhythms in locomotion and sleep-wake cycles regulated by PDYN neurons in the VMH (VMHPDYN+), we bilaterally injected AAV2/1-Ef1α-DIO-EGFP-p2A-mClk-Δ19 into the VMH of PDYN-Cre mice. Four weeks after viral expression, we obtained 24-hour locomotor activity and sleep recordings, using mice overexpressing dominant-negative CLOCKΔ19 (mClkΔ19) as the experimental group and EYFP-expressing mice (EYFP) as controls (Figures 1A and 1B). Although viral expression was primarily restricted to the VMH, a small number of EGFP-positive cells were occasionally observed in adjacent hypothalamic regions (Figure 1B, arrows). These off-target cells were sparse and did not appear to influence the behavioral results, as confirmed by the EYFP-only control group.”

In the Discussion of the manuscript (Page 9, paragraph 2, lines 1-8): “In the REM sleep cycle, we found that functional suppression of CLOCK activity of PDYN neurons in the VMH prolonged the latency of REM sleep and decreased the REM sleep cycles during the dark phase (Figure 5), indicating that maintaining a circadian rhythm is crucial for REM sleep homeostasis. Previous studies demonstrated that the circadian control of REM sleep is mediated by clock gene oscillations in the SCN [9, 41, 42]. Therefore, these findings raise the possibility that CLOCK activity in VMH-PDYN neurons may influence REM sleep through interactions with SCN circadian outputs or by functioning as an intrinsic regulatory node within the sleep-wake circuitry. CLOCKΔ19 is a dominant-negative mutant in which deletion of exon 19 abolishes CLOCK’s ability to transcriptionally activate E-box–dependent clock-controlled genes, while preserving heterodimerization with BMAL1. In homozygous mutant mice, this results in a lengthened circadian period that progressively transitions to arrhythmicity, highlighting the potency of CLOCKΔ19 as a functional inhibitor of circadian oscillator stability.”

Reviewer 2 Report

Comments and Suggestions for Authors

This interesting study addresses the relatively explored area via specifically targeting the Clock gene in Ventromedial Hypothalamic Prodynorphin Neurons expressing neurons, linking circadian mechanisms to REM sleep regulation. This finding advances understanding of the role of hypothalamus in sleep-wake cycles beyond general clock gene functions. Use of cell-type-specific AAV-mediated gene disruption in Cre mice, combined with polysomnography, locomotor monitoring, and EEG spectral analysis, provides comprehensive data. Statistical analyses are appropriate and well-reported. The authors found selective impairments in REM sleep (e.g., reduced duration, fragmentation, altered cycles) and circadian behaviors (e.g., reduced dark-phase activity within certain time windows, phase advances), supported by phase-specific analyses and spectral data. This highlights potential REM-specific roles, suggesting that VMH PDYN+ neurons may serve a node for circadian-REM integration, with broader relevance to sleep disorders, emotional regulation, and cognitive processes, opening avenues for targeted therapies.

Before recommending for publication, there are several crucial issues that should be addressed in revision:

  1. To improve clarity and accessibility (especially for readers outside neuroscience), please provide a complete "List of Abbreviations" section at the end of the paper or inline definitions upon first use. This should include expansions for all currently undefined terms (e.g., AAV, DMH, Csnk1e, EYFP, mClkΔ19, PDYN, REM/NREM, etc) and cover the entire manuscript.

  2. Add inline explanations for key terms like "Clock gene" or "gamma oscillations."

  3. Correct figure numbering inconsistencies: Figure 7 is referenced in the text but likely refer to Figure 6.

  4. Discussion would benefit from considering AAV targets beyond PDYN neurons, i.e. potential off-target effects (e.g., viral spread or incomplete specificity). Lack of additional validations (e.g., immunohistochemistry for Clock expression) can be discussed in limitations together with limited sample size (n=6-8 mice per group) and considerable variability (e.g., trends in rhythm indices, p=0.056) border on significance, risking false positives.

  5. As considered some effects can be circadian phase-dependent (or linked specifically to light vs. dark spans), but mechanisms (e.g., how Clock disruption could have altered EEG patterns or interplay with specific SCN areas) remain speculative. Lack of behavioral assays (e.g., anxiety, cognition) can be addressed among limitations.

  6. Limited generalability of results can be addressed as findings in male mice under controlled conditions may not fully extend to female rodents, other species, particularly diurnal like humans.

Author Response

Dear reviewer,

Thank you for your comments and professional advice. These opinions help to improve the academic rigor of our manuscript. Based on your suggestion and request, we have made the corrected modifications to the revised manuscript. We hope that our work can be improved again. Furthermore, we would like to show the details as follows:

Responses to comments from Reviewer #2

Q1: To improve clarity and accessibility (especially for readers outside neuroscience), please provide a complete “List of Abbreviations” section at the end of the paper or inline definitions upon first use. This should include expansions for all currently undefined terms (e.g., AAV, DMH, Csnk1e, EYFP, mClkΔ19, PDYN, REM/NREM, etc) and cover the entire manuscript.

A: We accept this comment. We have added a “List of Abbreviations” section at the end of the manuscript, ensuring all acronyms used are included.

List of Abbreviations

AAV

Adeno-Associated Virus

BMAL1

Brain and Muscle ARNT-Like 1

CLOCK

Circadian Locomotor Output Cycles Kaput

mClkΔ19

Mouse CLOCK mutant with a deletion of exon 19

Csnk1e

Casein kinase 1 epsilon

DMH

Dorsomedial Hypothalamus

EEG

Electroencephalogram

EMG

Electromyogram

EGFP

Enhanced Green Fluorescent Protein

EYFP

Enhanced Yellow Fluorescent Protein

NREM

Non-Rapid Eye Movement Sleep

P2A

Self-cleaving 2A peptide

PDYN

Prodynorphin

REM

Rapid Eye Movement Sleep

SCN

Suprachiasmatic Nucleus

VMH

Ventromedial Hypothalamus

VMHPDYN+

Ventromedial Hypothalamic PDYN-expressing neurons

DMH

Dorsomedial Hypothalamus

Per2

Period circadian regulator 2

Brn3a

Transcription factor Pou4f1

dMHb

Dorsal Medial Habenula

IRI

The inter-REM interval

Q2: Add inline explanations for key terms like "Clock gene" or "gamma oscillations."

A: Thank you for this helpful comment. we have checked and ensured that key terms such as “Clock gene and gamma oscillations” are provided with clear inline explanations upon first use.

As shown in the “Intoduction” of the manuscript (Page 2, lines 4-8): “The timing, duration, and continuity of REM sleep are tightly governed by the internal circadian clock, which is orchestrated by a core set of genes and proteins, including the central transcription factor Clock [8–11]. While disruption of the Clock gene and CLOCK protein has been shown to impair sleep architecture and circadian rhythms at the whole-brain level [12–14], its role within specific neuronal populations remains incompletely understood.”

As shown in the “2.5 EEG Power Normalization” of the manuscript (Page 4, lines 15-17): “For each frequency band of interest (delta (δ): 0.5-4 Hz, theta (θ): 6-10 Hz, sigma (σ): 12-16 Hz, gamma (γ): 30-60 Hz), the band power was computed as the integral of the power spectrum within the corresponding frequency range.”

Q3: Correct figure numbering inconsistencies: Figure 7 is referenced in the text but likely refer to Figure 6.

A: Thank you for catching this. We have thoroughly checked all figure references in the manuscript and confirmed that the reference to Figure 7 was indeed an error and has been revised to Figure 6. All inconsistent figure number references have been corrected to ensure that the figure legends accurately correspond to the text description.

As shown in the “Results” of the manuscript (Pages 8, lines 1-9), “In the light phase, mClkΔ19 mice showed a significant reduction in gamma energy during REM sleep compared with EYFP mice (F (3,48) = 7.658, p = 0.0003, Two-way ANOVA with Šídák’s multiple comparisons test, Gamma: p = 0.0011; Figure 6A), whereas no significant differences were observed during NREM sleep or wakefulness (Figures 6B and 6C). In the dark phase, gamma energy in mClkΔ19 mice was significantly decreased during both REM sleep and wakefulness, while delta energy during wakefulness was markedly increased relative to controls; no significant changes were detected during NREM sleep (REM: F(3,48) = 5.472, p = 0.0026, Gamma: p = 0.0048; WAKE: F (3,48) = 7.144, p = 0.0005, Delta: p = 0.048, Gamma: p = 0.002; NREM: F (3,48) = 0.089, p = 0.966, Two-way ANOVA with Šídák’s multiple comparisons test; Figures 6D-6F).”

Q4: Discussion would benefit from considering AAV targets beyond PDYN neurons, i.e. potential off-target effects (e.g., viral spread or incomplete specificity). Lack of additional validations (e.g., immunohistochemistry for Clock expression) can be discussed in limitations together with limited sample size (n=6-8 mice per group) and considerable variability (e.g., trends in rhythm indices, p=0.056) border on significance, risking false positives.

A: We accept this comment. We have added a discussion of this point in the “Discussion” section of the manuscript (Page 10, lines 31-36): “Additionally, we acknowledge that AAV delivery carries the possibility of some viral spread and off-target expression. In the description of the Methods and Figure 1, we emphasized the specificity of the VMH injection site and noted that despite potential off-target effects, the specific behavioral and physiological phenotypes we observed (e.g., reduced REM sleep and EEG changes) are highly consistent with literature reports of VMH dysfunction, strongly supporting the VMHPDYN+ circuit as the primary mediator.”

Q5: As considered some effects can be circadian phase-dependent (or linked specifically to light vs. dark spans), but mechanisms (e.g., how Clock disruption could have altered EEG patterns or interplay with specific SCN areas) remain speculative. Lack of behavioral assays (e.g., anxiety, cognition) can be addressed among limitations.

A: We accept this comment. We have made the following revisions and expansions in the “Discussion” section of the manuscript (Page 10, lines 18-22): “The precise mechanisms by which CLOCK interference alters the EEG spectrum, especially reduced REM gamma and increased wake delta, remain speculative. We need to integrate fiber photometry or calcium imaging to directly monitor the activity and rhythmic changes of VMHPDYN+ neurons.”

We have listed the lack of behavioral assays for anxiety, cognition, etc., as a limitation. Given the known association of EEG changes (REM gamma waves and wake delta waves) with cognitive and emotional states, we suggest that future research should prioritize the exploration of these behavioral phenotypes. We have made the following revisions and expansions in the “Discussion” section of the manuscript (Page 10, lines 22-30): “and investigate whether changes in the EEG frequency bands result in changes in cognitive levels by behavioral tests in mice.

Previous studies have found that the disruption of CLOCK can induce an elevated anxiety level and mania-like behavior [13, 56]. REM sleep is known to regulate emotional behaviors [4, 57–59]. Therefore, the observed reduction in REM sleep duration following CLOCK suppression in the VMHPDYN+ neurons may have significant implications for emotion-related behaviors in mice, a possibility that warrants systematic investigation.”

Q6: Limited generalability of results can be addressed as findings in male mice under controlled conditions may not fully extend to female rodents, other species, particularly diurnal like humans.

A: We accept this comment. We have made the following revisions and expansions in the Discussion: We point out that this study was conducted only in male mice and emphasize that the VMH region exhibits significant sexual dimorphism. Therefore, future studies should explore whether the function of the Clock gene in these neurons differs in female mice.

In the “Discussion” section the manuscript (Page 10, line 29-32): “Given the established sexual dimorphism in VMH circuitry [60–62], our future study should specifically explore the potential gender differences associated with this circadian rhythm regulation, thereby further enhancing the generalizability of the conclusions.”

Round 2

Reviewer 1 Report

Comments and Suggestions for Authors

Although proof of molecular clock disruption by overexpressing mtCLOCK would strengthen the manuscript, we understand the limitations mentioned by the authors and feel the added explanation is sufficient.  Overall, the authors did an excellent job addressing this reviewer's concerns.